# Effect of Complex Road Networks on Intensive Land Use in China's Beijing-Tianjin-Hebei Urban Agglomeration

**Chen Zeng [1,2,*], Zhe Zhao [3], Cheng Wen [4,5], Jing Yang [1] and Tianyu Lv [1]**

[1]  Department of Land Management, Huazhong Agricultural University, Wuhan 430070, China;
yangjing_ivy@webmail.hzau.edu.cn (J.Y.); tianyulv97@webmail.hzau.edu.cn (T.L.)

[2]  Institute of Geographical Sciences and Natural Resources Research, Chinese Academy of Sciences,
Beijing 100101, China

[3]  Department of Public Management, Remin University of China, Beijing 100872, China; zhaozhe@ruc.edu.cn

[4]  School of Geography, University of Leeds, Leeds LS2 9JT, UK; C.Wen@leeds.ac.uk

[5]  Research Institute of Environmental Law, Wuhan University, Wuhan 430072, China

*  Correspondence: zengchen@igsnrr.ac.cn; Tel.: +86-132-9668-3817

**Abstract:** Coupled with rapid urbanization and urban expansion, the spatial relationship between transportation development and land use has gained growing interest among researchers and policy makers. In this paper, a complex network model and land use intensity assessment were integrated into a spatial econometric model to explore the spatial spillover effect of the road network on intensive land use patterns in China's Beijing–Tianjin–Hebei (BTH) urban agglomeration. First, population density, point of interest (POI) density, and aggregation index were selected to measure land use intensity from social, physical, and ecological aspects. Then, the indicator of average degree (i.e., connections between counties) was used to measure the characteristics of the road network. Under the hypothesis that the road network functions in shaping land use patterns, a spatial econometric model with the road network embedded spatial weight matrix was established. Our results revealed that, while the land use intensity in the BTH urban agglomeration increased from 2010 to 2015, the road network became increasingly complex with greater spatial heterogeneity. The spatial lag coefficients of land use intensity were positively significant in both years and showed a declining trend. The spatially lagged effects of sector structure, fixed asset investment, and consumption were also significant in most of our spatial econometric models, and their contributions to the total spillover effect increased from 2010 to 2015. This study contributes to the literature by providing an innovative quantitative method to analyze the spatial spillover effect of the road network on intensive land use. We suggest that the spatial spillover effect of the road network could be strengthened in the urban–rural interface areas by improving accessibility and promoting population, resource, and technology flows.

**Keywords:** transportation; complex network; spatial econometric model; land use; POI

## 1. Introduction

Transportation development has transformed land use patterns worldwide in the past several decades [1]. Transportation construction essentially requires land, and transportation land is inherently an important built-up land use. Thus, transportation land is a non-negligible source of urban land growth, and transportation development is a crucial factor that could explain the ubiquitous urban sprawl or urban expansion phenomenon [2]. On the other hand, in fast-growing cities in developing countries, to some extent, transport development improves public transportation in the city center, helps facilitate livelihoods, and potentially prevents sprawl [3]. The unclear stimulation or control of

urban expansion through transportation construction causes ambiguity in the complicated relationship between transportation and land use. This work aims to explore this problem by using the BTH urban agglomeration in China, which is both the political center and most rapidly developed region, as the case study area.

China has a remarkably high growth rate of built-up land area, and transportation land expansion in major food production regions has increased considerably in the period of 2000–2010 [4]. Transportation development has also experienced remarkable changes in terms of intra-rearrangements and interconnections [5]. Intra-rearrangement of road accessibility and the expansion of public transportation helps facilitate the daily commute, alleviates traffic congestion, and promotes urban redevelopment [6], whereas strengthening interconnections among counties and cities are set as the benchmarks of advancing regional development, such as promoting the balanced and coordinated development of the BTH urban agglomeration [7]. Coupled with transportation development, residential, industrial, and commercial land and various development zones have rapidly sprawled to the suburban and urban–rural fringe zones in coastal megacities, such as Shanghai, Shenzhen, and Guangzhou, and extended toward medium-sized cities and megacities in the central China [8].

Given the limit of natural resources and ecological carrying capacity of megacities threatened by high-density anthropogenic activities, China's Ministry of Natural Resources and the 13th Five-year National Plan have emphasized compact and efficient land use, as well as balanced regional development for controlling the scale of industrial enterprises in cities, towns, and homesteads in rural areas [1,9]. A series of intensive land use assessment projects have been promoted in different counties and cities to promote the concept of land use intensity and improve land use efficiency [10]. The idea of land use intensity first emerged in the agricultural domain to examine the relationship among the inputs of seed, fertilizer, mechanization, cropland, and the outputs of yields and production. In response to the massive urban land expansion, intensive land use has gradually embraced the concepts of compact development, high density, and mixed land use with variety and vitality [11]. The achievement of these concepts requires urban redevelopment or urban renewal that can alleviate pressures related to the development of city centers and urban expansion in the urban fringes. On the other hand, land use efficiency refers to the ability to achieve maximum output under the conditions of a given investment on land. Greater land use efficiency is usually related to greater land use intensity. In China, the incremental land use planning on the outward land expansion used to be the focus. However, while urban sprawl has been widely discussed and critically controlled, focus has moved to inventory land use planning, which refer to the planning on the extant land resources to improve land use efficiency. To explore the potential of extant urban land resources and achieve efficient land use, land use intensity assessment and spatial optimization through both mixed and clustering land use pattern are widely implemented [12].

This study uses China's BTH urban agglomeration, as the case study area to examine how complex road networks affected land use intensity between 2010 and 2015. To answer this question, an original quantitative assessment framework to link complex road networks with intensive land use is provided. The road network is embedded into a spatial spillover model to measure its influence on land use intensity. It is declared that we focus on the effect of road network on land use intensity, and the influence of land use on road development is not considered in our study. The outline of the paper is presented as follows. A comprehensive literature review is presented in Section 2 to justify the proposed research framework and elaborate on the relationship between inter-/intra transportation development and inward/outward land use. Material is described in Section 3. The spatial modeling approach is explained and the spillover effect of the road network on intensive land use is explored in Section 4. Section 5 provides the results of the road network estimation and spatial models. The results, and our conclusions are presented in Section 6. The policy implications and future recommendations of the study are presented in Section 7.

## 2. State of the Art and Literature Review

The integration of transportation development and intensive land use for optimized resource allocation has captured the interest of scholars, planners, and public authorities. Empirical land use and transportation interaction models have been established to investigate the driving forces behind transportation development in urban expansion. Integrated land use-transportation planning has been utilized to support the coordinated urban–rural development, balance the intra- and interurban development, and meet the targets of Intergovernmental Panel on Climate Change for reducing carbon emissions [13]. However, the interface between transportation systems and intensive land use requires further exploratory investigation and empirical studies to formulate a systematic framework for achieving sustainable development.

Figure 1 illustrates our research framework which takes account of the relationship between regional transport and land use, wherein transport construction requires land space, while transport development has spatial spillover effect on land. The spatial spillover effect, which means the attributes in one observation would be influenced by the attribute in the neighboring observations, is a popular term in regional development research. The concept of the spatial spillover effect in the transportation network was proposed in the late 1990s and has been applied to a series of studies on regional socioeconomic development [2]. In the realm of regional development, both intra-transport and inter-transport are simultaneously promoted in the context of China's New Urbanization Strategy. Intra-transport patterns lead to land transformation through providing sufficient pedestrian and bicycle space, as well as strengthening metro system and public transportation. In promoting inter-transport, trains, air and ship transportation have been expanding with great magnitude. In this process, intra-transport is coherently correlated with infill development and inter-transport provides the essential prerequisite of outward development [2,5]. On the other hand, in China, land spatial planning has been initiated to integrate the original land use planning and urban planning since 2018. In the contemporary spatial planning, the infill development facilities the program of urban redevelopment and the outward development is the primary form of urban expansion [11,12]. Empirical studies have justified the close relationships between intra-transport and infill development, as well as the close relationship between inter-transport and outward development in mega-cities and urban agglomerations [1,2]. However, whether inter-transport also drives the infill development and whether there is spatial spillover effect during this process remain unsettled [2].

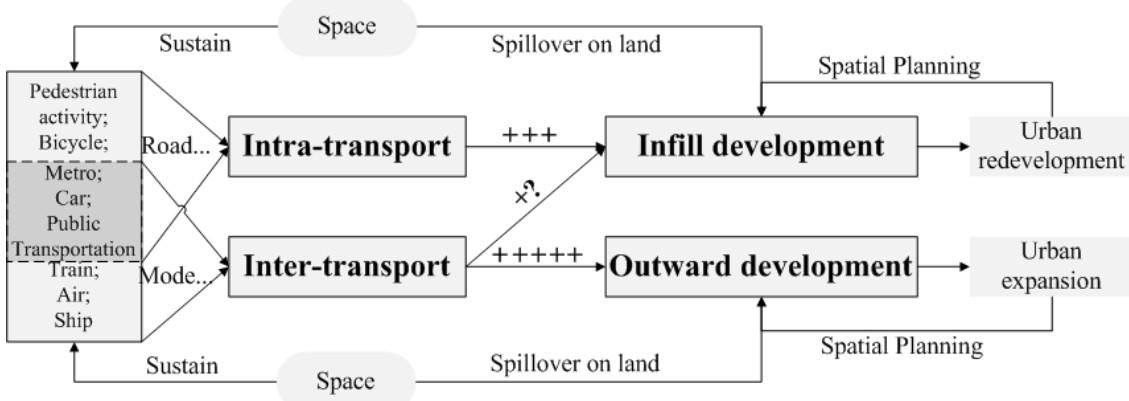

**Figure 1.** Regional transportation development and land use.

## 2.1. Intensive Land Use versus Outward Expansion

China has undergone widespread urban land expansion due to its unprecedented urbanization and socioeconomic development [12]. Empirical studies have proven that excessive artificial land expansion leads to consequent increases in greenhouse gas emissions, urban heat effects, air pollution, ecosystem degradation, and various eco-environmental issues [9]; these problems have attracted considerable attention from the academia and policymakers [13]. Furthermore, rapid economic development largely depends on land-related revenues, whereas economic output considerably relies on land inputs that raise the issue of land use efficiency [14]. In this context, similar to the concepts of smart growth and compact cities, land use intensity is proposed as a means to control urban land expansion and improve land use efficiency in China.

Generally, the measurement of land use intensity often focuses on land use density, land use structure and spatial pattern; these approaches are generally divided into single and multi-index groups. Scholars have technically used population density, economic output per land, and other indicators describing landscape patterns to measure land use intensity [12]. For example, the intensity of urban land use is measured by combining population and economic densities [15]. Conceptual framework of Erb et al. [16] integrated three dimensions: (a) input intensity, (b) output intensity, and (c) the associated system-level impacts of land-based production [16] in measuring land use intensity. Yang et al. later extended the third dimension from the landscape perspective [17]. From the perspective of the input intensity, indicators are needed to reflect the degree of function-input of the land [18]. In output intensity dimension, the indicator of population or economic revenue are often selected to demonstrate the capacity of the land [17]. With regard to landscape dimension, indicators such as the aggregation can be employed to measure the spatial pattern of land use [17]. In China, national specifications for built-up land use intensity assessment have been issued (TD/T 1018-2008). Land use intensity assessments in development zones (TD/T 1029-2010) include macroscopic socio-economic indicators (such as gross domestic product (GDP) per hectare, fixed assessment investment per hectare, and built-up land per capita) and micro indicators (such as floor ratio of residential land and land transaction revenue per hectare).The land use intensity assessment is performed on industrial land using the indicators of the layout, scale, internal structure, and economic intensity of the industrial enterprises [19]. At present, assessments of land use intensity have been widely applied to different land use types at counties, cities, and provinces of different regions [20,21]. Intensive land use programs and related appraisal movements are strongly promoted in China to regulate the scale of built-up land and achieve sustainable land development [14,20]. In China, land use has gradually transformed from the outward mode to compact patterns [1]. As the counterpart of outward expansion, intensive land use is considered an impelling tool for transforming incremental land use planning into inventory land use planning that has been promoted to facilitate urban redevelopment in several cities.

Based on theories of land science and urban and regional economy, land use intensity is affected by various factors with temporal and spatial heterogeneity. In urban areas, the infrastructure, economy, and market in land use intensity or urban land efficiency have a considerable positive influence in the medium-sized and large cities (2007–2015) [22]. Economic factors, such as gross domestic product and tertiary sector proportion, also have a remarkable positive influence on Wuhan urban agglomeration in China [17]. Moreover, technological externalities and the "threshold effect" with respect to the factors of intensive land use also exist. Peng et al. found that a diversified economy helps to improve intensive land use and this effect functions through technological externalities when the city population is above 546,000 [23].

## 2.2. Transportation and Land Use

Transportation and land use are an integrated system in the spatial domain. The optimization of their relationship is a challenge and opportunity among transport planners, land use planners, engineers, environmentalists, and urban and regional developers [13,24]. Although extensive research has been performed on the integration of transportation systems and land use systems, as well as

the relation between these concepts, studies on how to address the problems caused by the spatial planning and coordination of transportation facilities and land use remains limited [1,25].

The influence of transportation on land use has been examined at the local and regional scales with respect to land development, land value, and integrated spatial planning [24]. Transportation land development is inherently determined by the development of railway and highway systems [26]. Road network is the prerequisite of population and capital flow, as well as the expansion of diversified sectors, projects, and markets, which keeps shaping land use patterns. To characterize the rapidly developed transportation network, the complex network model is widely applied in transportation; the topological graph is the prerequisite of the model which is achieved by simplifying lines and intersections as edges and nodes [27]. In addition to describing the transportation network, there is a series of studies on how the development of large-scale transportation infrastructures, such as major road construction, has transformed the land cover in specific regions. Transit-oriented development is remarkably effective in balancing the supply driven by transportation and the demand of land; thus, the integration of transportation and land use has been advocated [25]. Furthermore, the high level of transit accessibility increases residential land use intensity by changing the residential location and commute mode choice. An important reflection of transit-oriented development is the land value changes due to variations in the transportation construction [24]. Transportation construction enhances the accessibility and amenities of neighborhoods and increases the land value. Land revenues are arguably a primary source of the local budget. Public transportation projects in cities of developing countries, such as Delhi and Ho Chi Minh, are funded and implemented through the "Land-for-Infrastructure" mechanism which utilizes and captures land use value [28]. In the meantime, land value is considered during the feasibility analysis in transportation projects to prevent excessive land use through an economic model based on the relationship between the elasticity of land price and the estimated future value of land. The bid-rent land use model is effective in locating jobs and population. Land use limits were set to analyze a delineated traffic zone based on the transportation system capacity [29]. These studies proves the spillover effect of transportation on land use, which to some extent, provides theoretical support for treating transportation network as the medium to generate the spatial spillover effect on intensive land use.

Consequently, the spatial interactions caused by transportation at different spatial scales (such as village, county, and city) are stimulated to influence regional development by planners and policy-makers [28]. Transportation networks can influence the spatial structure of socioeconomic development by improving the population flow and commodity transfer in the spatial domain [30]. It has been widely acknowledged that the spatial spillover effect of the transportation network exist, which is also the hypothesis of our study [2]. Empirical evidence on the spillover and diffusion effect of road networks was first found in developed countries with advanced automobile and infrastructure development. The spatial influence of transportation networks on urbanization, productivity, trade, investment, and environment has then been widely investigated and verified in several developing countries [31]. The relationship between the transportation network and the administrative boundary with institutional hierarchy has also been explored.

Although the spatial spillover effect of transportation networks has been empirically confirmed, the magnitude and mechanisms of this effect differ in various transportation modes, areas, and time periods [2]. Studies on highways and railways, which are major sources of spatial spillover effects, have highlighted the important role of the former in promoting regional development over the last several decades [5]. The spatial heterogeneity of this effect has also been examined. For example, apparent disparities in the magnitude of the spillover effect have been found in the middle, eastern and western areas in China [32]. Practically, the spatial model was widely used to analyze the spillover effect across regions when a certain attribute value between regional spatial unit and neighborhood spatial unit has high probability of spatial correlation. There are three types of spatial econometric models: spatial autoregressive model (SAR), spatial error model (SEM), and spatial Durbin model [33]. Spatial autoregressive model can capture the influence of spatially lagged independent

variables. In the spatial error model, the spillover effect exists in the error of the disturbance term. The spatial Dubin model (SDM) combines the spatial influence of both the dependent variable and the independent variable from the neighbors. However, the application of these spatial models in combining transportation and land use remains rare. Hence, additional empirical studies are expected to further quantitatively measure the spatial spillover effect of transportation development on land use in specific fast-growing urban agglomerations.

## 3. Materials

As China's core political and cultural region, the BTH urban agglomeration has an area of approximately 216,000 km². It consists of 15 cities, namely, Beijing, Baoding, Cangzhou, Chengde, Handan, Hengshui, Langfang, Shijiazhuang, Tianjin, Tangshan, Xingtai, and Zhangjiakou (Figure 2). In 2017, the population of the BTH urban agglomeration reached 110 million, and its GDP totaled 8300 billion RMB. In China, BTH is one of the most vibrant urban agglomerations with clustered high-educated workforce and investments. It also represents an important pilot area for innovation-driven regional development. In China's New Urbanization Plan (2014–2020), BTH aims to become the global urban agglomeration. The infrastructure in this area is superior to that in other areas of China, and its transportation network is complex with functional nodes and grids as specified in the Integrated Transportation Planning for the Cooperative Development in the BTH urban agglomeration [34].

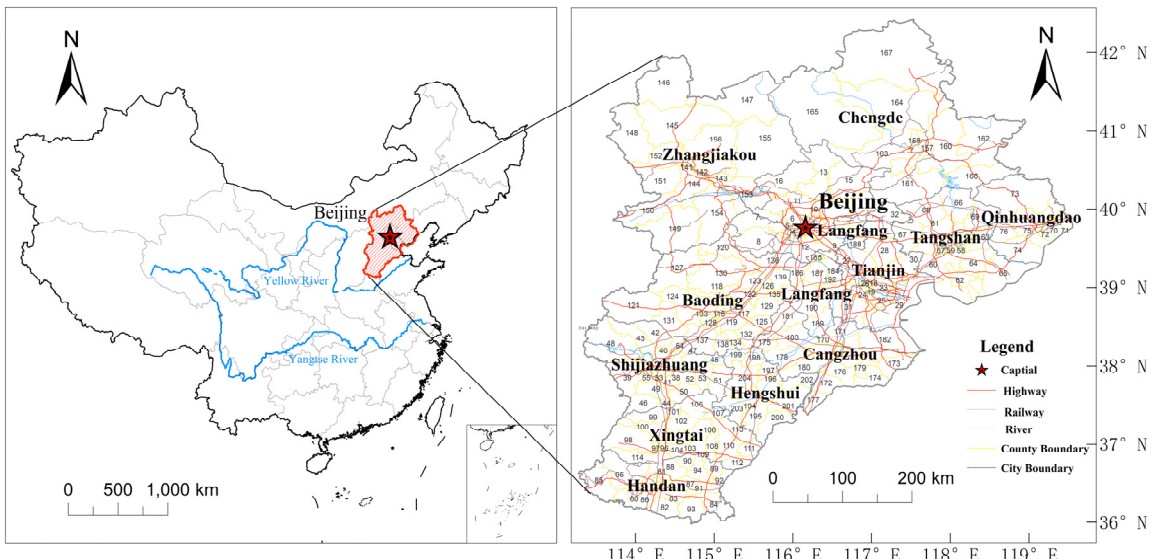

**Figure 2.** Study area.

Table 1 lists data collected and sources for this study, including land use classification image, the spatial distribution of road network, point of interest (POI) data, administrative division data and socioeconomic data of Beijing, Tianjin, and Hebei in 2010 and 2015. The land use classification product was retrieved from the Geographical Information Monitoring Cloud Platform which was supported by the Institute of Geographic Sciences and Natural Resources Research, Chinese Academy of Sciences. The road distribution and administrative division dataset were obtained from Geographical Information Monitoring Cloud Platform, which provided a number of free open source dataset [2]. POIs were retrieved from the Baidu API platform to include the sectors of food, hotels, shopping malls, education and training, transport facilities, finance, real estate, and corporate business [12]. Based on the literature review, we also collected socio–economic datasets, namely, population, GDP, urbanization, sector structure, investment, consumption [17,35], from the Statistical Yearbooks of Beijing, the Statistical Yearbooks of Tianjin, and the Statistical Yearbooks of Hebei Province in 2010 and

2015. These indicators are stored as the potential influencing factors to explain the change on land use intensity in Beijing–Tianjin–Hebei urban agglomeration.

**Table 1.** Data source overview.

| Data | Data Type | Data Source |
|---|---|---|
| Land use classification data (Interpreted from Landsat TM/ETM images in 2010 and 2015 (spatial resolution of 30 m)) | Cropland, grassland, forest, build-up land, water and others | Geographical Information Monitoring Cloud Platform (http://www.dsac.cn/DataProduct) |
| Road network data | Road | Geographical Information Monitoring Cloud Platform (http://www.dsac.cn/DataProduct/Detail/201843) |
| Point of interest (POI) data | The sectors of food, shopping malls, education and training, transport facilities, finance, real estate, and corporate business | Baidu API platform (http://lbsyun.baidu.com/index.php?title=lbscloud) |
| Administrative division dataset | Provincial boundaries, city boundaries and county boundaries | Map World in National Platform for Common Geospatial Information Services (https://www.tianditu.gov.cn/) |
| Socio-economic dataset | Population, GDP, urbanization, sector structure, investment and consumption | the Statistical Yearbooks of Beijing, the Statistical Yearbooks of Tianjin, and the Statistical Yearbooks of Hebei Province in 2010 and 2015 |

## 4. Methodology

A complex network model and land use intensity assessment was integrated into spatial econometric model to explore the spillover effect of the road network on intensive land use patterns in the BTH urban agglomeration. The measurement of land use intensity, construction of the complex road network model, and the ultimate spatial models are described in detail in the following sub-sections.

### 4.1. Measurement of Land Use Intensity

In this paper, the assessment of land use intensity was mainly based on the conceptual framework of land use intensity constructed by Erb et al. [16], and also referred to the extension of the dimensional connotation of this framework by Yang et al. [17]. There were three indicators to measure land use intensity, namely population density, POI density, and aggregation index. POI density was chosen because POIs signify an important built environment attribute of a compact urban form and represent the tangible inputs that human activities on urban land produce [18]. As a result, we have extracted the POIs of the sectors of food, shopping malls, education and training, transport facilities, finance, real estate, and corporate business. Meanwhile, aggregation index is widely implemented to measure spatial heterogeneity and is inclined to reflect the overall degree of clustering and coherence of landscape spatial pattern [17]. Hence, we calculated aggregation index for each county or district. Then, the information entropy method, based on the principles of extracting useful information with the applied data, was used to calculate the relative weights of population density, POI density and aggregation index [36]. According to the degree of variation of the indicators, the weight value of each indicator can be calculated objectively, providing a more reliable basis for the comprehensive evaluation of multiple indicators [37]. Ultimately, the land use intensity was calculated as the weighted sum of these indices (Equation (1)).

$$\text{LUI}_i = I_{i\_PD} \times W_{PD} + I_{i\_AI} \times W_{AI} + I_{i\_POID} \times W_{POID} \tag{1}$$

where LUI$_i$ is the land use intensity for each administrative unit; I$_{i\_PD}$, I$_{i\_AI}$, and I$_{i\_POID}$ are the normalized values of the population density, aggregation index, and POI density of each administrative unit, respectively; W$_{PD}$, W$_{AI}$, and W$_{POID}$ are the weights of these three indices, respectively. Aggregation index was calculated by Fragstats software, and the number of POIs for each county was obtained by the spatial join function of ESRI ArcGIS 10.1.

### 4.2. Construction of Road Complex Network

The complex network model is widely applied in transportation related studies, and several approaches may be used to represent the transportation network [38]. In the study, the L-space method was used to quantify and visualize the road network in the BTH urban agglomeration [38]. The L-space method was used due to its advantages in demonstrating the topology of the road network by assigning road intersections to nodes and considering two adjacent nodes as an edge [27]. On the basis of the accessibility features of the road network, the intersections of the road network were treated as nodes, and the roads were treated as edges to formulate the complex network model here. Two hypotheses were also proposed: (1) The road network was regarded as the unidirectional network model, in which if intersection A is accessible to the intersection B through the edge, then B is also accessible to A with the same edge. (2) The road network was regarded as an unweighted network model. The topological structure and feature were quantified and analyzed through a series of indicators through the software of ESRI ARCGIS 10.1 and Pajeck 5.08. Average degree, network radius, and average edge number are commonly used indicators to measure network characteristics. In this paper, the average degree of the road network was used as the indicator to formulate the spatial interaction relationship. In general, degree refers to the number of connections for each node. In the complex network, average degree is calculated as the average value of the total degrees and reflects the compactness of the intersections. The larger the value is, the more compact the intersections are in the network (Equation (2)).

$$\bar{d} = \frac{1}{N} \sum_i k_i \tag{2}$$

where $\bar{d}$ is the average degree while $N$ and $k_i$ are the numbers of intersections and edges linking all the intersections, respectively.

### 4.3. Selection of Socioeconomic Explanatory Variables

In this paper, considering regional characteristics and data accessibility, nine potential factors were selected, which are urbanization rate (UR), population density (PD), gross domestic product per capita (PGDP), proportion of industrial sector (PIS), proportion of tertiary sector (PTI), total social consumption per capita (PSSC), fixed asset investment (FAI), disposable personal income for rural residents (DPIR), and disposable personal income for urban residents (DPIU). All these factors are the common socio-economic statistics and are retrieved from the Statistical Yearbook in 2010–2015. Correlation tests between these nine factors and land use intensity were used to eliminate factors that have no correlation with land intensity and those showing high levels of multicollinearity. Ultimately, three explanatory variables were selected, namely the proportion of tertiary sector (PTI), fixed asset investment per land (PFAI), and total social consumption per capita (PSSC) to examine their influence on land use intensity. The proportion of tertiary sector (PTI) represents the proportion of the added value of the tertiary industry to the gross domestic product. The correlation coefficients between PTI and LUI are 0.6253 and 0.7232, in 2010 and 2015 respectively. Fixed asset investment per land (PFAI) represents the fixed asset investment amount per square kilometer of land. It is an indicator to measure the economic benefits of land use [39]. The correlation coefficients between PFAI and LUI are 0.7046 and 0.7146, in 2010 and 2015 respectively. Total social consumption per capita (PSSC) is a direct indicator of consumption demand, representing the amount of goods sold by society [2]. The correlation coefficients between PTI and LUI are 0.5662 and 0.6850, in 2010 and 2015 respectively. These three

variables were selected since their correlation coefficients are all above 0.5. In addition, tertiary sector development occupied a considerable position in the sector structure in the Beijing–Tianjin–Hebei urban agglomeration and PTI was found to largely influence the spatial allocation of land resource [40]; secondly, retail sector development was one of the key sectors giving rise to increasing transformation of land use into commercial and recreational uses; lastly, A series of pilot projects with large fixed asset investment were carried out in Beijing–Tianjin–Hebei urban agglomeration, which gave the explanation on its prominent role in influencing land use intensity.

### 4.4. Complex Spatial Model Based On Road Network

In this paper, a spatial interaction modes based on the road network characteristics is established to characterize the spatial spillover effect of the intra-transport in counties or cities on regional development. The interaction matrix among these county or city units is formulated on the basis of the gravity model to quantitatively evaluate the strength of interaction and determine the relative accessibility degree in the regional road network. The gravity model is generally used in representing the gravity force between two objects and is widely applied to measure the degree of spatial interaction [17]. As presented in Equation (3), the attribute value $d$ is positively correlated with the estimated gravity value $Gij$, whereas distance $Dij$ has a negative relationship. The spatial influence of neighboring counties through the road network is based on the gravity model as follows:

$$Gij = \frac{\overline{d_i} \times \overline{d_j}}{(Dij)^2} \tag{3}$$

where $Gij$ is the interaction between county $i$ and $j$; $\overline{d_i}$ and $\overline{d_j}$ are the average degrees of county $i$ and $j$ respectively, and $Dij$ represents the geographic distance between county $i$ and $j$.

Then $Gij$ was used to construct the spatial weight matrix. The spatial econometric models was used to explore the spillover effect of the road network. Based on literature and our preliminary data analysis, the general spatial econometric model was specified as follows:

$$P = \beta_0 + \alpha W(F_{ij})P' + \sum_{i=1}^{m} \beta_i x_i + \sum_{i=1}^{m} W(F_{ij})\beta_j x_j + \gamma W(F_{ij})_3 \varepsilon \tag{4}$$

where $p$ is the value of land use intensity in each county; $\beta_0$ refers to the constant term; $\beta$ is the coefficient for the explanatory variable; $x$ denotes the explanatory variables, which are the explanatory variables of proportion of tertiary sector (PTI), fixed asset investment per land (PFAI), and total social consumption per capita (PSSC), respectively.

The general spatial econometric model form is specified as Equation (4). The spatial weight matrix is given as W which is based on the reciprocal value of the average degree calculated through Equation (3). $\alpha$ is the spatial lag coefficient, $P'$ is the spatial neighbors of the observation; $\gamma$ is the spatial coefficient of the error term; When $\gamma = 0$ and $\beta_j = 0$, Equation (4) turns out to be SAM (spatial autoregressive model); When $\alpha = 0$ and $\beta_j = 0$, Equation (4) turns out to be SEM (spatial error model); When $\gamma = 0$, Equation (4) turns out to be SDM (spatial Durbin model). Moran's I statistics and Lagrange tests were also used to evaluate SAR, SEM, and SDM.

## 5. Results

The results of spatial-temporal change of land use intensity, the road network characteristics, the influencing factors and the spatial spillover effect of road network in Beijing–Tianjin–Hebei urban agglomeration are presented as follows.

*5.1. The Spatio-Temporal Change of Land Use Intensity*

The ultimate land use intensity values exhibited spatial and temporal changes between 2010 and in 2015 in the BTH urban agglomeration. The weights of POPD, POID, and AI generated by the entropy method were 0.36, 0.31, and 0.34 in 2010 and 0.33 and 0.34 in 2015, respectively. The descriptive statistics of land use intensity in each county or district were calculated as specified in Equation (1) (Table 2).

**Table 2.** The descriptive statistics of land use intensity in the BTH urban agglomeration.

|  | LUI | | |
|---|---|---|---|
|  | **2010** | **2015** | **Change Ratio** |
| Mean | 0.4274 | 0.5133 | +20.10% |
| Standard deviation | 0.0288 | 0.0206 | −28.47% |
| Maximum value | 0.9980 | 0.9448 | −5.33% |
| Minimum value | 0.0337 | 0.2358 | +599.70% |
| Range | 0.9643 | 0.7090 | −26.48% |

In general, the mean values of land use intensity in the BTH urban agglomeration increased from 0.4274 to 0.5133, but the gap between minimum and maximum values declined from 0.9643 to 0.7090. The standard deviation of land use intensity also decreased by 28.47%. These results show that the average level of land use intensity in the urban agglomeration improved, while the gap between counties and districts narrowed between 2010 and 2015.

Figure 3A presents the spatial distribution of land use intensity in the BTH urban agglomeration in 2015 and Figure 3B illustrates the spatial distribution of the change rate of land use intensity in the BTH urban agglomeration from 2010 to 2015. In 2015, high values clustered in the southeastern area and low values were clustered in the northwestern area. Most of the districts in Beijing and Tianjin showed high land use intensity values, especially in Dongcheng, Xicheng, and Chaoyang Districts in Beijing and in Heping, Nankai, and Hexi Districts in Tianjin. Rural areas of Zhangjia Kou and Cheng De Prefectures in Hebei Province showed the lowest levels. From 2010 to 2015, the northwest and west of the BTH urban agglomeration showed the largest land use change rates, which were the northern districts of Beijing, the western and northern counties of Chengde, and the central and eastern regions of Zhangjiakou in the northwest of the BTH urban agglomeration, as well as the western part of Baoding City located in the west of the BTH urban agglomeration. The spatial effects of Beijing and Tianjin influenced nearby administrative units, such as Langfang, Cangzhou, and Tangshan, where land use patterns became substantially more intensive.

Figure 4 illustrates the hotspot pattern of land use intensity in the BTH urban agglomeration in 2010 and 2015. Hot spots indicate statistically significant high-value clusters of land use intensity, while cold spots indicate statistically significant low-value clusters of land use intensity. Both in 2010 and in 2015, hot spots were mainly concentrated in Beijing, Tianjin and Langfang areas, while Zhangjiakou and Chengde were clusters of cold spots. The hotspots of land use intensity in all counties in Hengshui and Qinhuangdao were not significant in both years. There were three specific changes from 2010 to 2015. First, from perspective of hot spots, the range of significant hotspot districts in Beijing was expanding, and the significance of most districts is becoming higher. Among them, Chaoyang District leaped from a significant level of 90% to 99%, showing the greatest change. Tianjin continues to show a highly significant (99%) cluster of hot spots, except for Jixian County. Second, with respect to cold spots, it can be observed that Zhangjiakou and Chengde showed a shrinking trend in term of the range of cold spots from 2010 to 2015. In Zhangjiakou, the significant levels of cold spots in all counties except Xuanhua have been reduced, whereas in Chengde, Fengning is the only county whose significance of cold spots has changed from a significant to insignificant. Third, most of the counties in the southern regions of the BTH urban agglomeration have insignificant hotspots of land use intensity. Only Jinzhou County in Shijiazhuang maintained a 99% significant level of hotspot clustering from 2010 to 2015.

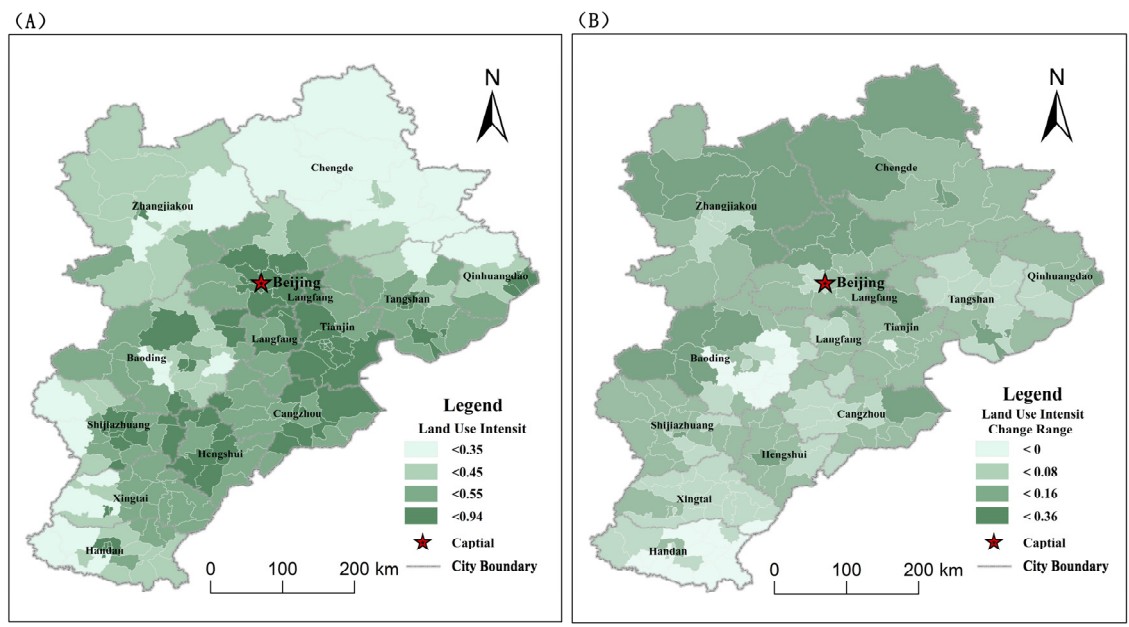

**Figure 3.** (**A**) Spatial distribution of land use intensity in the BTH urban agglomeration in 2015; (**B**) Spatial distribution of the change rate of land use intensity in the BTH urban agglomeration from 2010 to 2015.

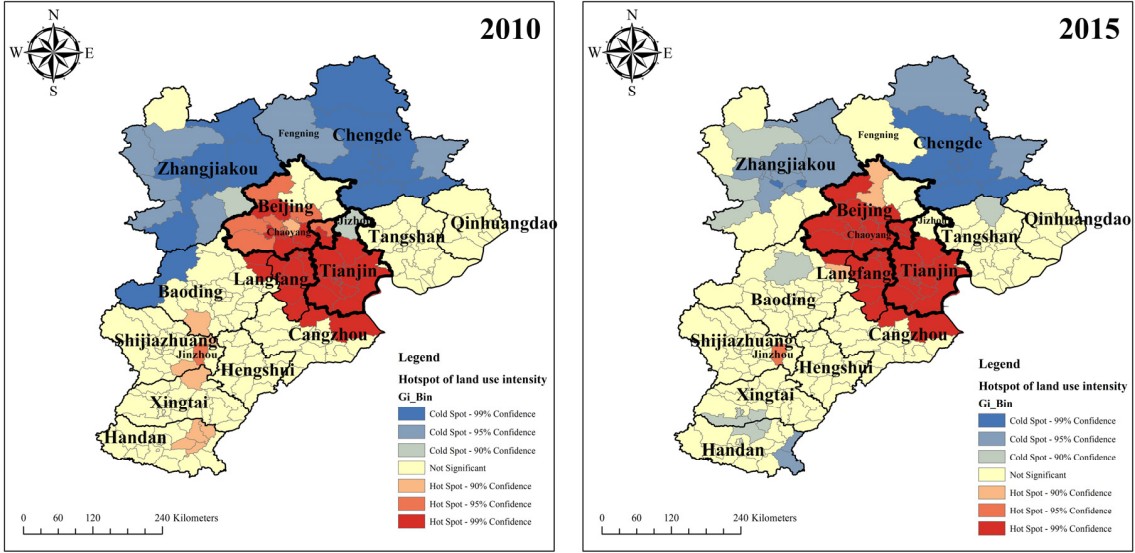

**Figure 4.** Hotspot pattern of land use intensity in the BTH urban agglomeration in 2010 and 2015.

*5.2. Complex Road Network Characteristics*

Table 3 presents the 2010–2015 statistics of the highways running through this agglomeration. The highway mileage increased from 192,300 km to 225,003 km with a growth rate of over 17% from 2010 to 2015. The growth rate of the expressway is higher than those of the arterial and secondary highways. The road density increased from 0.89 km/km$^2$ to 1.03 km/km$^2$ from 2010 to 2015. Moreover, the scale of the transportation network expanded rapidly with the trans-boundary expressway contributing a large share of development.

**Table 3.** Transportation network construction in the JingJinJi urban agglomeration.

| | | Beijing | Tianjin | Hebei | Total |
|---|---|---|---|---|---|
| Highway mileage (km) | 2010 | 21,114 | 14,832 | 154,344 | 192,300 |
| | 2015 | 21,885 | 16,550 | 184,553 | 225,003 |
| | Growth rate | 3.65% | 11.58% | 19.57% | 17.01% |
| Expressway (km) | 2010 | 903 | 982 | 4307 | 8202 |
| | 2015 | 982 | 1130 | 6333 | 10,460 |
| | Growth rate | 8.75% | 15.07% | 47.04% | 27.53% |
| Arterial highway (km) | 2010 | 924 | 1040 | 4307 | 8281 |
| | 2015 | 1393 | 1260 | 5408 | 10,076 |
| | Growth rate | 50.76% | 21.15% | 25.56% | 21.68% |
| Secondary highway (km) | 2010 | 3196 | 3165 | 15,872 | 24,243 |
| | 2015 | 3361 | 3224 | 19,656 | 28,256 |
| | Growth rate | 5.16% | 1.86% | 23.84% | 16.55% |
| Road density (km/km$^2$) | 2010 | 1.29 | 1.30 | 0.82 | 0.89 |
| | 2015 | 1.33 | 1.39 | 0.97 | 1.03 |
| | Growth rate | 3.66% | 6.95% | 18.08% | 15.47% |

Figure 5 illustrates the degree and the corresponding number of nodes in 2010 and 2015 and Figure 6 illustrates the topological structure and average degree of the complex road network (Figure 6A in 2010 and Figure 6B in 2015). In 2010, the road network has 1513 nodes and 4904 edges, and in 2015, the numbers have improved to 1623 nodes and 5299 edges. In both 2010 and 2015, there are the highest numbers of nodes at the degree level of 3, followed by the degree level of 4. That is to say, a large proportion of nodes in road complex have 3 or 4 edges to connect. There were the least number of nodes with the degree larger than 6. From 2010 to 2015, the number of nodes with the degree of 3 and 4 also increased remarkably, whereas nodes with a degree of 2 declined. The average degree of the complex road network rose from 3.24 to 3.27, which means that the accessibility in the urban agglomeration was enhanced. From the spatial distribution pattern of average degree (Figure 5), an unbalanced pattern of transportation construction was observed in Beijing, Tianjin and Shijiazhuang. The highest value was mostly clustered in the city center, particularly the urban district in Beijing and Tianjin in both years. The average degree did not show substantial increase in Beijing, but considerable growth in nodes with degrees greater than 3 was observed in Tianjin and Hebei. However, the proportion of the nodes with the degree between 3 and 4 in Hebei remained behind that of Beijing and Tianjin, indicating that Beijing and Tianjin showed a better topological structure and higher accessibility than Hebei.

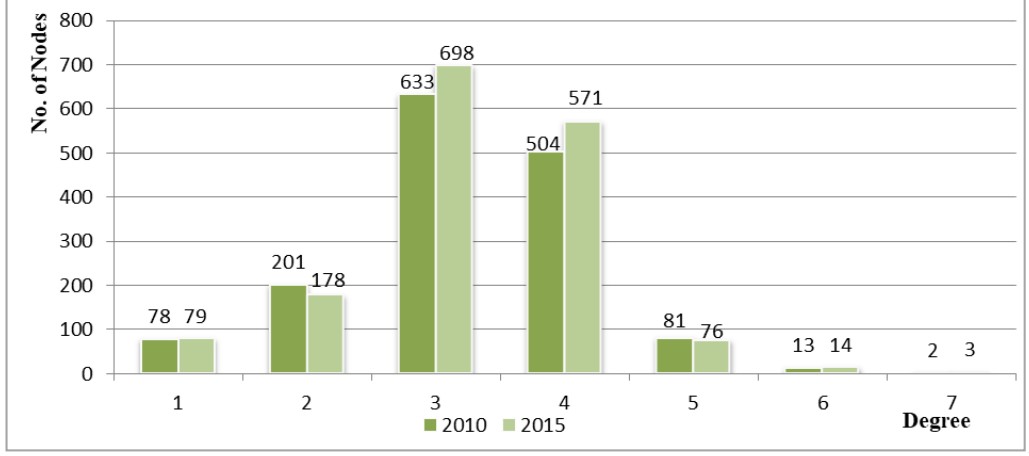

**Figure 5.** Network characteristics of degree in BHT urban agglomeration.

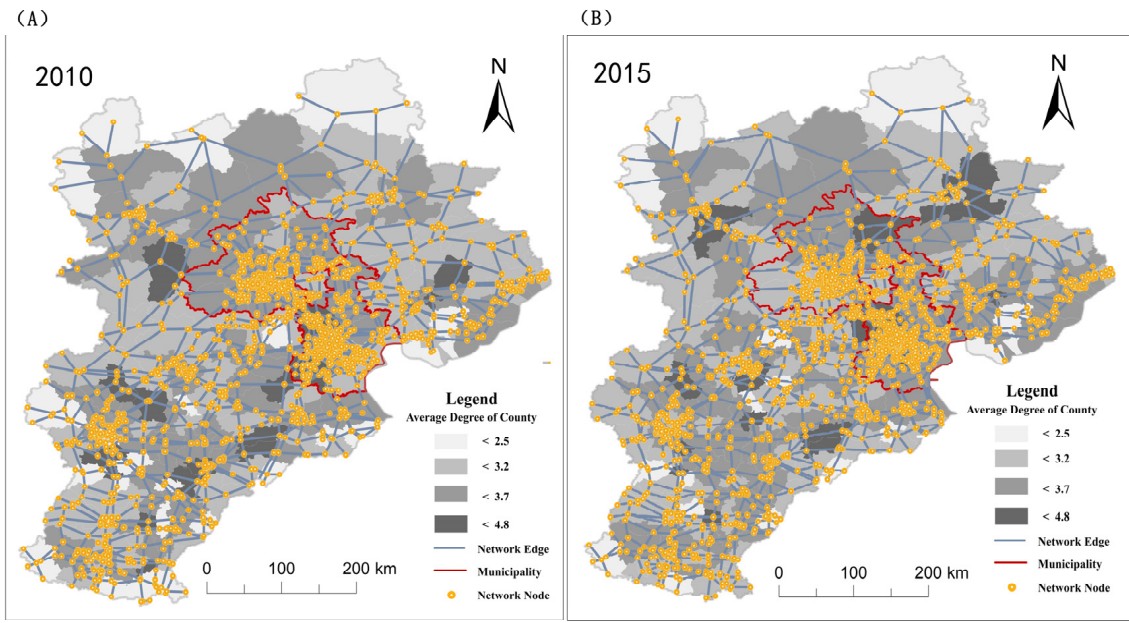

**Figure 6.** Road network in the BTH urban agglomeration.

*5.3. Spatial Spillover Effect on Intensive Land Use through Road Network*

The linear regression model by ordinary least square (OLS) algorithm and two spatial econometric models were implemented, which are SAR and SDM, to examine the spatial lag effects on land use intensity through the road network and check spatial spillover effects from the explanatory variables PTI, PFAI, and PSSC. The diagnoses of the spatial autocorrelation tests and significant spatial auto-correlations (*p* value at 0.01 significance level) were observed in 2010 and 2015 with respect to land use intensity in the BTH urban agglomeration. According to the Lagrange tests, it was shown that SAR performed better than SEM, as a result, we applied SAR and SDM models to present the spatial econometric model results.

The results of the spatial econometric models were shown in Table 4. The influences were significant in the direct spatial lag term, i.e., the spatial lag of the independent variable, and error term in 2010 and 2015 (*p* value at the significance level of 0.01). From 2010 to 2015, the spatial lag coefficient decreased from 0.4966 to 0.0314 greatly but still remained statistically significant. A similar declining trend also occurred in the coefficient of the spatial error term. In 2010, the correlation coefficients of PTI and PSSC were larger whereas that of PFAI was smaller than the results produced by ordinary least square algorithm. These results indicated that the contribution of PFAI declined whereas the contributions of PTI and PSSC strengthened under the influence of spatial lag. In 2015, the factors with increasing correlation coefficients compared with the results of OLS also included PTI and PSSC. From 2010 to 2015, the contributions of PTI and PFAI to land use intensity declined (except PTI in SEM), whereas the influence of PSSC increased in OLS and SAR.

**Table 4.** Spatial regression results.

| Year | 2010 | | | 2015 | | |
|---|---|---|---|---|---|---|
| Model | OLS | Model1_ SAR | Model2_SDM | OLS | Model1_ SAR | Model2_SDM |
| $R^2$ | 0.5670 | 0.5930 | 0.8908 | 0.6518 | 0.6740 | 0.9735 |
| PTI | 0.2493 *** | 0.2190 *** | 0.4715 *** | 0.2320 *** | 0.2208 *** | 0.2237 *** |
| PFAI | 0.0637 *** | 0.0519 *** | 0.1998 *** | 0.0452 *** | 0.0363 *** | 0.0264 *** |
| PSSC | $2 \times 10^{-6}$ ** | $2 \times 10^{-6}$ *** | $1 \times 10^{-6}$ | $3 \times 10^{-6}$ *** | $4 \times 10^{-6}$ *** | $3 \times 10^{-6}$ *** |
| W_ PTI | | | −0.2758 *** | | | −0.0278 |
| W_ PFAI | | | −0.0020 | | | −0.0121 |
| W_ PSSC | | | $-2 \times 10^{-7}$ | | | $-3 \times 10^{-6}$ *** |
| ρ | | 0.0789 *** | 0.0683 *** | | 0.0399 *** | 0.7802 *** |
| λ | | | | | | |
| Log likelihood | | 163.8480 | | | 226.7947 | |

Notes: PTI is the proportion of the tertiary sector, PFAI is fixed asset investment per land, PSSC is the total social consumption per capita, ρ is the spatial lag coefficient in the spatial lag model, λ is the spatial error coefficient in the spatial error model. *** refers to the significance level at 0.01, ** refers to the significance level at 0.05.

In SDM, the spatial lag coefficients (ρ) were 0.9289 and 0.9154 in 2010 and 2015, respectively, at the significance level of 0.01. The correlation coefficient of PTI became insignificant, while the correlation coefficient of PFAI became less significant. The positive influences of PSSC were strengthened by threefold. The spatially lagged influences of PTI (W_ PTI) and PSSC (W_ PSSC) changed from positive to negative, whereas the coefficient of the spatially lagged PFAI (W_ PSSC) dramatically increased from $2 \times 10^{-5}$ to 0.2809. In general, the contributions of spatially lagged variables weakened, but the local influences strengthened in 2010–2015, which was largely attributed to the influences of PSSC.

In terms of model performance, SDM showed superiority over SAR and SEM with higher R-square values, which indicated that spatial spillover effects of the road network can explain land use intensity.

## 6. Discussion

This work investigates the effect of the transportation network on land use intensity and explores the spatial spillover effect by treating the road network as a medium influencing land use patterns. The BTH urban agglomeration is taken as the case study area integrating interdisciplinary approaches of multi-index assessment, complex network analysis, and a spatial econometric model to investigate the driving mechanism of intensive land use.

The ambiguity in the relationship between the transportation network and land use intensity motivates this study. Transportation development is an important driver of land use change in both developed countries like the United States and developing countries like China. Land use change can be both outward and infilling. The results exhibit that the indicators of transportation network such as degree and edge numbers have increased, and so have the values of land use intensity in BTH urban agglomeration. It is shown that a number of new roads have been built or extended in the districts and counties, which increases the degree of the transportation network, helps the infill development and diffuse the pressure in the city center. Transportation becomes a vital prerequisite to achieve coordinated regional development as well as intensified land use. In fact, China has experienced an era in which regional development was strongly emphasized as several mega-cities underwent social and eco-environmental problems due to high-density population and economic activities in the city center. The development of transportation infrastructure has been considered by the government as an important tool to promote regional balance. The Beijing–Tianjin–Hebei urban agglomeration is a typical area for the implementation of regional development plan. This is because Beijing is the capital city of China, Tianjin is the municipality directly under the central government and Hebei is the normal province, and there are huge differences in terms of socio-economic development. As revealed by this study, although the spatial spillover effect of road network in the Beijing–Tianjin–Hebei urban agglomeration declined from 2010–2015, the degree of road network in Hebei Province which

lags behind Beijing and Tianjin has increased noticeably. The regional land use intensity have also grown in most administrative units, which reflects the influence of road network on the infill land use development.

Technically, this study contributes to the methodology for measuring the role of transportation network and determining whether it helps improve land use intensity at the regional level. Instead of generating a clear-cut positive or negative coefficient of the road network on intensive land use, transportation network is treated as a medium that produces the spatial spillover effect on regional land use change. When the spatial spillover effect of the road network on regional intensive land use is discussed, the "distance" should embody the essential function of roads as the flows of human and materials are largely based on transportation "roads".

The spatial spillovers realize their interaction through the road network. This phenomenon responds to the idea of treating transportation network as the medium to function in the spatial modeling of land use intensity. It may be different from the previous studies on the spatial spillover effect of the transportation network [41]. The characteristics of road network is used generate the gravity force and then is embedded into the spatial weight matrix in spatial econometric models. The results of spatial modeling based on the complex road network model help to reveal the path of this process and provide insights for regional development. For example, in 2010 and 2015, the spatially lagged variables prove to have significant impact on land use intensity with declining spatial influence, while fixed asset investment in the neighboring administrative units greatly influenced the local land use intensity with increasing power through the road network. In this situation, fixed asset investment in neighbors with satisfactory road linkages is an important factor in shaping the local land use pattern when spatial land use planning is formulated in certain regions of the Beijing–Tianjin–Hebei urban agglomeration.

This study has two limitations largely due to data availability. (1) The transport network refers to road network. Railways were not included because subway and metro, intercity-train, high-speed rail have their featured spatial spillover effects which are of great difference with that in road. Meanwhile, several railway projects are currently under construction, and their operation will take time to implement. We do not have the complete dataset of railways. Thus, the arterial and secondary road networks are focused on to investigate their influence and enhance the accuracy and specificity of the results. This selection was made due to its ability to reflect the network characteristics and accommodate the data features. (2) The road network was considered as an unweighted network model, which means we do not consider the traffic flow and length of the road. The unweighted network model is a classical model in the field of complex network research, which has been widely used in traffic network congestion evacuations [42], traffic robustness analysis [43], and urban planning. In recent years, the weighted network model based on road levels has been proposed and became fashionable in complex network research [44]. In this paper, due to the data availability, we adopt the unweighted network model to analyze the road network topology.

As railway system has experienced unprecedented development in China, it is expected to include railways in the transportation network analysis in the future. The different types of transport can be of different weights when they are embedded into the spatial econometric model given a necessary dataset. This differentiated treatment is useful because a defined relationship between land use and transportation hardly exists, especially at the regional level. The influences of socioeconomic development on land use could be realized through the use of transportation channels for the flows of population and goods.

## 7. Conclusions, Limitation and Policy Implication

### 7.1. Conclusions

The spatial spillover effect of the road network on intensive land use in 200 counties and districts in the Beijing–Tianjin–Hebei urban agglomeration is explored by using a spatial econometric model integrated with a complex network model, landscape ecology indicators, and multi-index assessment. The results revealed that the spatial spillover effects on land use intensity through road network were positively significant with a declining trend although land use intensity increased in the urban agglomeration between 2010 and 2015 and the road network became increasingly advanced with intense spatial heterogeneity. The results indicating the relationship between land use and transportation confirmed that inter-transportation construction could help promote intensive land use. However, the contribution declined in Jing-jin-ji in the period of 2010–2015, and rapid urban expansion through transportation development was implied. The spatially lagged effects of the industrial sector structure, fixed asset investment, and consumption were significant in the majority of our spatial econometric models. The increasing contributions to land use intensity from 2010 to 2015 reflected the spatial spillover effect of road networks on land use intensity and the changing influences of different factors. These results implied that road network development helped to produce the spatial spillover effect by improving accessibility and promoting population, resource, and technology flows. As core cities of the urban agglomeration, Beijing and Tianjin have demonstrated a radiating effect on the surrounding areas with respect to intensive land use patterns. However, the less developed economy and infrastructure of Hebei had low resource allocation efficiency due to their inferior context of labor division and industrial development.

### 7.2. Policy Implication

In the future, it is recommended the strengthening of transportation networks to shape land use patterns and promote coordinated development in the integrated Beijing–Tianjin–Hebei urban agglomeration. Several policy suggestions are listed below.

(1) In the spatial land use plan of the Beijing–Tianjin–Hebei urban agglomeration, the spatial spillover effect on the transportation network should be considered to improve land use intensity, especially in the urban–rural transition area. The spatial spillover effect through road network on land use intensity is an important indicator on whether transportation promotes or hinders urban expansion. According to the result of significant and declining spatial spillover effect of road network on intensive land use, transportation development should be promoted in the form of integrated transportation and land use development to strengthen the spatial interaction comprehensively. The coordinated population–land–industry development is encouraged in the peripheral urban–rural counties such as Fangshanin Beijing, Jixian in Tianjin, and Xuanhua in Zhangjiakou in Hebei Province to guide the transportation development that will be implemented in an intensive manner [18].

(2) The transportation network in the urban agglomeration should be improved in a systematical way by enhancing the road topological structures of Beijing and Tianjin and promoting road network construction in Hebei to narrow the gap among Beijing, Tianjin, and Hebei. The density and depth of important nodes in the road network should be increased in the central areas of Beijing and Tianjin to maintain the spatial influences on land use intensity through transportation network. New roads are expected to appear in the northern area in Hebei province. Moreover, the accessibility and compactness of road networks should be improved in the urban–rural interface to provide a basis for generating the spatial spillover effects in the urban agglomeration.

(3) Transportation and land use plan should be assessed and strengthened in the urban–rural interface through the holistic investigation on the socioeconomic development in these areas. The influence of transportation on land use intensity is dependent on its function in urban–rural transition areas. In counties such as Miyun in Beijing, Jixian in Tianjin and most counties in Chengde city in Hebei province, the interactions between transportation and land use were weak due to their

physical environment and their low levels of socio-economic development. Our results confirm the contribution of sector structure, investment, and consumption to regional land use intensity. Depopulation in rural areas due to the emigration to urban areas and rural public infrastructure development are also important influencing factors of the regional intensive land use and urban–rural transformation [45]. As a result, positive interactions between transportation and efficient land use require better coordination among the social, economic, cultural, and technological development in the context of urban–rural transformation.

**Author Contributions:** Conceptualization, C.Z.; methodology, C.Z. and Z.Z.; software, J.Y.; validation, C.W. and Z.Z.; formal analysis, T.L.; investigation, T.L.; resources, C.Z.; data curation, Z.Z.; writing, C.Z. and C.W.; writing—review and editing, C.W.; visualization, J.Y.; supervision, C.Z.; project administration, C.Z.; funding acquisition, C.Z. All authors have read and agreed to the published version of the manuscript.

**Funding:** This research was funded by National Natural Science Foundation of China, grant number 41771563 and Chinese Postdoctoral Foundation, 2019T12013.

**Acknowledgments:** The authors would like to thank the anonymous reviewers for their valuable comments.

**Conflicts of Interest:** The authors declare no conflict of interest.

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
