# Peer review of "Effect of Complex Road Networks on Intensive Land Use in China’s Beijing-Tianjin-Hebei Urban Agglomeration"

_land, doi:10.3390/land9120532_

Round 1

Reviewer 1 Report

The paper is well structured. In detail describe carried scientific research and its results. It is based mainly on known methodology, which is tested through case project (urban agglomeration in China). Like authors recognised, it will be better to have projection of transport flow on presented road links and also included all other modes of transport in model, primary rail transport which could make significant changes in presented results.

Also, the paper should reconsider some minor changes and improvements:

  • There is a very large number of “References” which should be shortened (some of them cover one sentence/one general expression).
  • The reference number 70 is mentioned in text but miss on the list of references
  • In the paper is often used “study” and should be written “paper”
  • In the paper authors in some parts write “we”. All paper should be written in third face singular.
  • Figure 1 is should be described in detail.
  • Under every title should be some texts with explanation. It is missing under title for chapter 5.
  • Need to correct some tipfelers in texts (red number; title which is on the end of the page, one row of table 3 is on the end of the page and other part of the table is on next page…)
  • Under title 8 Patents should delete first general sentence which is actually input for authors.

Author Response

The paper is well structured. In detail describe carried scientific research and its results. It is based mainly on known methodology, which is tested through case project (urban agglomeration in China). Like authors recognised, it will be better to have projection of transport flow on presented road links and also included all other modes of transport in model, primary rail transport which could make significant changes in presented results.

Thank you for your comment. It is indeed a good idea to include all other transport mode in our model, the rail transport in particular. However, after implementing the data collection and analyzing again, we find it difficult to incorporate the influence of rail in the extant model. It might be better to have another paper to continue this study. The detailed explanation is also presented in Section 7.2.

This study has two limitations largely due to data availability:

1) The transport network refers to road network. Railways were not included in this study because subway and metro, intercity-train, high-speed rail have their featured spatial spillover effects which are of great difference with that in road. Meanwhile, several railway projects are currently under construction, and their operation will take time to implement. We do not have the complete dataset of railways. Thus, the arterial and secondary road networks are focused to investigate their influence and enhance the accuracy and specificity of the results. This selection was made due to its ability to reflect the network characteristics and accommodate the data features.

2) The road network was considered as an unweighted network model, which means we do not consider the traffic flow and length of the road. Unweighted network model is a classical model in the field of complex network research, which has been widely used in traffic network congestion evacuations (Zhang et al., 2019), traffic robustness analysis (Shang et al., 2020) and urban planning. In recent years, weighted network model based on road levels has been proposed and became fashionable in complex network research (Ding et al., 2019; Tian et al., 2016). In this paper, due to the data availability, we adopt the unweighted network model to analyze the road network topology.

As railway system has experienced unprecedented development in China, it is expected to include railways in the transportation network analysis in the future. The different types of transport can be of different weights when they are embedded into the spatial econometric model given necessary dataset. This differentiated treatment is useful because a defined relationship between land use and transportation hardly exists, especially at the regional level. The influences of socioeconomic development on land use could be realized through the use of transportation channels for the flows of population and goods.

Also, the paper should reconsider some minor changes and improvements:

  • There is a very large number of “References” which should be shortened (some of them cover one sentence/one general expression).

We have shortened the number of “references” to 46.

  • The reference number 70 is mentioned in text but miss on the list of references

We have relisted the reference and the original number 70 has been deleted.

  • In the paper is often used “study” and should be written “paper”

We have changed the term “study” into “paper”.

  • In the paper authors in some parts write “we”. All paper should be written in third face singular.

We have made revisions and all the descriptions are written in third face singular.

  • Figure 1 is should be described in detail.

Thank you for your comment. We have added a paragraph to provide the detailed descriptions on Figure 1 as follows:

Figure 1 illustrates our research framework to take account of the relationship between regional transport and land use, wherein transport construction requires land space, while transport development has spatial spillover effect on land. In the realm of regional development, both intra-transport and inter-transport are simultaneously promoted in the context of China’s New Urbanization Strategy. Intra-transport patterns lead to land transformation through providing sufficient pedestrian and bicycle space, as well as strengthening metro system and public transportation. Private cars are expanding greatly and car-based commuting mode has been increasingly popularized in metro-, large-scale, and medium-scale cities. In promoting inter-transport, trains, air and ship transportation have been expanding with great magnitude. In this process, intra-transport is coherently correlated with infill development and inter-transport provides the essential prerequisite of outward development. On the other hand, in China, the Land Spatial Planning has been initiated to integrate the original land use planning and urban planning since 2018. In the contemporary spatial planning, the infill development facilities the program of urban redevelopment and the outward development is the primary form of urban expansion. Empirical studies have justified the close relationships between intra-transport and infill development, as well as the close relationship between inter-transport and outward development in mega-cities and urban agglomerations. However, whether inter-transport also drives the infill development and whether there is spatial spillover effect during this process remain unsettled.

  • Under every title should be some texts with explanation. It is missing under title for chapter 5.

We have added the general explanation under title for chapter 5 as follows:

Based on the land use intensity assessment framework, complex road network and spatial econometric models, we have produced the results of spatial-temporal change of land use intensity, the road network characteristics, the influencing factors and the spatial spillover effect of road network in Beijing-Tianjin-Hebei urban agglomeration.

  • Need to correct some tipfelers in texts (red number; title which is on the end of the page, one row of table 3 is on the end of the page and other part of the table is on next page…)

We have carefully checked the manuscript again and correct all the tipfelers in texts.

  • Under title 8 Patents should delete first general sentence which is actually input for authors.

We have deleted first general sentence under title 8 Patents.

Reviewer 2 Report

The paper entitled “Effect of complex road networks on intensive land use in China’s Beijing-Tianjin-Hebei urban agglomeration” presents a comparative analysis of different spatial econometric models that describes the influence of infrastructures on land-use change. The issue is of high relevance for the scientific community since (as author highlighted in their work) although exist a common agreement of the fact that infrastructures are derived of land use transformations, there is no evidence of how, where and in what ways the two issues are related.

The paper tryes to construct its model, which is somehow innovative but has several limitations (provided here and there in the main text randomly).

To what concern a general impression, it is expected that an analysis like that reveals much more in detail the spatial distribution and contextualization of the indicators, providing in-depth evaluations and shoving detailed maps of gradients to see how space put conditions to the interaction between infrastructures and land use. Instead, this paper remains more at the comparative, general stage instead of on using typical space-evident tools such as hotspots, cluster analysis, Principal Component and so on.

To what concern specific points:

1) this work is written in poor English. Poor sentencing often reduces the argumentative discourse while limiting analytical comprehension;

2) the structure has to be revised: introductive statements are present here and there, the methodology is obscure in large parts, indicators are not presented adequately to check their reliability, sources of data are often missing;

3) Discussion: you measured 200 counties: it is expected that you make a spatial analysis of distribution in a detailed manner to see how local conditions produce different effects (so policies are site-specific and not general and vague)

To what concern specific comments, please check the attached file.

Good luck!

Author Response

We have provided the response to the comments in the attached responding letter. The revisions on the detailed comment in the manuscript are also made and can be found in the revised version of the manuscript. 

Round 2

Reviewer 2 Report

Although i see that Authors works diffusely to revise this work I still see that the manuscript need further revision because:

1) You logical sentencing remains too poor: too repetition and low clarity of the significance of certain phrases is still present (i pointed out all these passages in my detailed comments in the attached file)

2) Your introduction is often repetitive and bad structured. Please revise accordingly

3) Avoid long repetitive statements and cut the paper around your own work without pleonastic statements. This will prompt the clarity.

Best wishes.

Author Response

Thank you very much for the detailed comment in the manuscript. We have made revisions point by point in the revised version. Regarding the specific comment below, we have responded as follows.

Although i see that Authors works diffusely to revise this work I still see that the manuscript need further revision because:

1) You logical sentencing remains too poor: too repetition and low clarity of the significance of certain phrases is still present (i pointed out all these passages in my detailed comments in the attached file)

Thank you very much for your detailed comments. We have deleted or revised the repetitive phrases accordingly.

2) Your introduction is often repetitive and bad structured. Please revise accordingly

Thank you for pointing out our problem. We have deleted or revised the introductory sentences, or moved then to the introduction section accordingly.

3) Avoid long repetitive statements and cut the paper around your own work without pleonastic statements. This will prompt the clarity.

Thank you for your reminding. We have adjusted and shortened our sentences accordingly. The long repetitive statements have been cut. Hopefully, the revised version is of better clarity.